# Experimental Study on Coaxial Waterjet-Assisted Laser Scanning Machining of Nickel-Based Special Alloy

**DOI:** 10.3390/mi14030641

**Published:** 2023-03-12

**Authors:** Jiajia Wang, Bin Wang, Chenhu Yuan, Aibing Yu, Wenwu Zhang, Liyuan Sheng

**Affiliations:** 1Faculty of Mechanical Engineering & Mechanics, Ningbo University, Ningbo 315211, China; 2Ningbo Institute of Materials Technology & Engineering, Chinese Academy of Sciences, Ningbo 315201, China; 3PKU-HKUST Shen Zhen-Hong Kong Institution, Shenzhen 518057, China

**Keywords:** waterjet assisted laser, nickel-based special alloy, laser drilling, orthogonal experiment

## Abstract

The problems of the recast layer, oxide layer, and heat-affected zone (HAZ) in conventional laser machining seriously impact material properties. Coaxial waterjet-assisted laser scanning machining (CWALSM) can reduce the conduction and accumulation of heat in laser machining by the high specific heat capacity of water and can realize the machining of nickel-based special alloy with almost no thermal damage. With the developed experimental setup, the laser ablation threshold and drilling experiments of the K4002 nickel-based special alloy were carried out. The effects of various factors on the thermal damage thickness were studied with an orthogonal experiment. Experimental results have indicated that the ablation threshold of K4002 nickel-based special alloy by a single pulse is 4.15 J/cm^2^. The orthogonal experiment results have shown that the effects of each factor on the thermal damage thickness are in the order of laser pulse frequency, waterjet speed, pulse overlap rate, laser pulse energy, and focal plane position. When the laser pulse energy is 0.21 mJ, the laser pulse frequency is 1 kHz, the pulse overlap is 55%, the focal plane position is 1 mm, and the waterjet speed is 6.98 m/s, no thermal damage machining can be achieved. In addition, a comparative experiment with laser drilling in the air was carried out under the same conditions. The results have shown that compared with laser machining in the air, the thermal damage thickness of CWALSM is smaller than 1 μm, and the hole taper is reduced by 106%. There is no accumulation and burr around the hole entrance, and the thermal damage thickness range is 0–0.996 μm. Furthermore, the thermal damage thickness range of laser machining in the air is 0.499–2.394 μm. It has also been found that the thermal damage thickness is greatest at the entrance to the hole, decreasing as the distance from the entrance increases.

## 1. Introduction

In modern aero and industrial gas turbine engines, the combustion temperatures are increasing to optimize power and efficiency [1,2]. Nickel-based alloys have been widely applied in the hot end parts of aero and gas turbine engines because of their excellent high-temperature strength, fatigue properties, corrosion, and oxidation resistance [3]. The turbine blade is the first hot end part to be transferred by heat from the combustion chamber [4]. With the increased working temperature in front of the turbine, the system has higher and higher requirements for the hot end parts to withstand extreme service conditions such as high temperature, high pressure, and high speed. To improve the reliability and lifespan of these parts, holes are often drilled to allow air to pass through the internal passages to facilitate cooling. The diameter of these holes is generally less than 1.2 mm, and there are higher quality requirements for the side walls of the holes [5]. At present, the most commonly used drilling methods in the industry include electrical discharge machining (EDM), electrochemical machining (ECM), and laser beam machining (LBM) [6]. However, there are certain limitations to these techniques. The material versatility of EDM is poor, and the choice of electrode type and material directly affects the material removal rate and surface integrity of EDM [7]. The surface quality of ECM drilling is high without a recast layer, but the dimensional accuracy of the hole is difficult to control [8], and the system is complicated and expensive. LBM is one of the most widely used non-contact advanced machining techniques, with good material versatility, high machining efficiency, precise control of laser energy input, and flexibility of laser beam delivery. However, there are common defects in thermal machining methods such as a recast layer, an oxide layer, HAZ, and microcrack in LBM, which will reduce the performance and life span of the parts. Ultrafast laser machining technologies such as picoseconds and femtoseconds can reduce these defects, but their material removal efficiency is low, making it difficult to meet the needs of industrial production [9,10].

Due to the advantages of water recycling, no pollution, and low price, water-assisted laser machining technology has become a research hotspot. In addition, water has a high specific heat capacity. Under the same temperature rise, the heat absorbed by water is much greater than that of the target material. The remaining laser energy in the target material is mainly absorbed by fluid water during heat conduction and then taken away [11]. Therefore, water-assisted laser machining can avoid excessive oxidation, thermal stress, HAZ, and other damage during machining. Water-assisted laser machining methods mainly include underwater laser machining [12], waterjet-guided laser machining [13], and waterjet-assisted laser machining [14]. Underwater laser machining is the simplest and most commonly used method of water-assisted laser machining. Zhu et al. [12] found that laser ablation of solid media in water can significantly increase the ablation rate. Lu et al. [15] studied the mechanism of underwater laser drilling and proved that underwater laser drilling could greatly improve machining efficiency and surface quality. Tangwarodomnukun et al. [16] found that underwater laser machining of silicon can obtain a smaller recast layer and HAZ compared with the air. Krstulović et al. [17] found that underwater laser machining of aluminum can obtain higher machining quality and material removal rate than in the air. However, the bubbles and debris produced in the liquid will affect the machining efficiency [18]. In addition, the almost zero water speed over the workpiece surface causes less cooling effect and less potential to reduce HAZ. In order to avoid these effects, Kalyanasundaram et al. [19] designed a non-coaxial waterjet coupling cavity and carried out experimental research on coaxial and non-coaxial waterjet laser machining of brittle materials. Tangwarodomnukun et al. [14,20] proposed a paraxial waterjet-assisted laser machining technology, which heats and softens the target material by laser, and expels the softened material by waterjet. A clean and smooth cutting surface was obtained. However, the overall machining efficiency of this method is limited by the inability to control the thickness of the water layer on the workpiece surface. Subsequently, Tangwarodomnukun et al. [21] proposed a new laser ablation technique. In this technique, a waterjet is used to impact the workpiece surface to create a thin and flowing layer of water. Experiments have found that this technique minimizes redeposition and HAZ. However, there is still a HAZ with a thickness of about 20 μm. Chree et al. [22] proposed to enhance the underwater laser ablation process by controlling the water flow rate and flow direction in a closed water chamber. By performing laser ablation on single crystal silicon in flowing water, the influence of air bubbles, water waves and debris on the laser beam can be reduced, and better machining quality can be obtained than that in still water. However, there are still some microcracks and micropores on the cutting surface.

Waterjet-guided laser machining technology uses a coaxial micro-waterjet as the medium to guide the total internal reflection of the laser beam. This method has the advantages of high surface quality, small recast layer, high precision, and long working distance [23]. It is currently the best machining quality in water-assisted laser machining technology. SYNOVA and Avonisys have commercialized the technology. It has been used in aerospace, semiconductor, medical, and other fields. However, the waterjet diameter limits the resolution of waterjet-guided laser machining. The essence of the waterjet-guided laser machining principle is to use the laser to remove materials. When facing the machining requirements of high-quality, high-depth-diameter/width-ratio holes, grooves, edges, and narrow working spaces, waterjet tends to lose its laminar flow state, which affects laser beam transmission [24]. At the same time, due to the multiple total internal reflections of the laser beam in the waterjet, the laser energy loss caused by stimulated Raman scattering (SRS) is relatively large. In addition, the waterjet is disturbed during machining due to the splashing of debris. Moreover, waterjet-guided laser machining equipment is generally expensive and has high maintenance costs, which limits the application of this technology in the manufacturing industry [25]. To avoid these problems, Madhukar and Mullick et al. [26,27,28,29,30,31] proposed a coaxial waterjet-assisted laser machining method. This method is different from waterjet-guided laser machining. The laser beam passes through the optical window and focuses on the surface of the target material without total internal reflection in the waterjet. It has the advantages of smaller HAZ and fewer microcracks. However, the millisecond laser with a wavelength of 1.07 μm has a large energy loss in water, and the laser energy required to remove a unit volume of material is higher. Wang et al. [32] proposed an immersion waterjet-assisted laser micromachining technology, using a 527 nm nanosecond pulse laser with less energy attenuation in water, and verified through experiments that the machining quality of this technology is better than that in air. Liu [11] conducted an experimental study on coaxial waterjet-assisted laser film cooling hole machining of nickel-based single-crystal superalloy. The microstructure of the sidewall of the film cooling hole was studied by transmission electron microscope (TEM), and it was found that the thickness of the heat-affected zone decreased with the increase of the waterjet speed.

Compared with laser machining in air, water-assisted laser machining does have many advantages. Thus far, the research on water-assisted laser machining is mostly related to the grooving, cutting, and micromachining, and there are few reports on water-assisted laser drilling of nickel-based alloys. The primary method of water-assisted laser drilling is percussion drilling, which has a limited drilling diameter and low precision, requiring secondary machining. Since water-assisted laser machining needs to rely on the conventional mechanical moving platform, which is difficult to realize the machining of holes with complex three-dimensional structures. In addition, the inclined waterjet has a weak flushing effect on the hole. With the increase of the drilling depth, the flushing effect of the waterjet on the ablated materials is weakened, which affects the drilling efficiency. For nanosecond laser, because of the slow motion of the mechanical axis, the laser pulse overlap is high, further aggravating the thermal effect. Therefore, this paper proposes a coaxial waterjet-assisted laser scanning machining (CWALSM) method. The laser beam and high-speed waterjet have impinged on the workpiece surface machining area in a perpendicular direction. By adding a scanning galvanometer, the laser beam scans and removes materials within the high-speed and large-diameter waterjet range. The nanosecond pulse laser is selected to improve machining efficiency and reduce thermal damage. This paper mainly focuses on the effects of laser pulse energy, laser pulse frequency, pulse overlap, focal plane position, and waterjet speed on the thermal damage thickness, which is studied by orthogonal experiment. Furthermore, the machining ability of CWALSM on nickel-based special alloy was researched experimentally.

## 2. Laser-Waterjet Coupling Characteristics

### 2.1. Mechanisms of CWALSM

Different from underwater laser machining (Figure 1a) and waterjet-assisted laser machining (Figure 1b), by configuring the waterjet coaxially with the laser beam (Figure 1c), when the waterjet with a certain pressure hits the target, the ablated material and cavitation bubbles in the machining area are taken away by the waterjet. At the same time, the cooling effect of water can effectively reduce the temperature of the material heated by the laser and reduce the conduction of heat to the surrounding materials and the thermal stress imbalance caused by the heating of the material. CWALSM is one of the water-assisted laser machining, as shown in Figure 2. The laser beam passes through the quartz glass and is focused on the workpiece surface after being transmitted in the laminar waterjet. The scanning galvanometer controls the high-speed scanning movement of the laser beam in the radial range of the waterjet diameter. Therefore, the diameter of the waterjet needs to be larger than that of the hole being machined. According to the different machining apertures, the diameter of the water column is generally ø2~ø5 mm. It can significantly reduce the coupling difficulty of laser and waterjet, and avoid the laser beam from damaging the nozzle wall due to mechanical vibration and other reasons, thereby improving system stability.

Water-assisted laser machining involves the interaction of three factors: laser, water, and target material, which is more complicated than laser machining in the air. In a gas environment, when a high-energy short-pulse laser beam irradiates the surface of the target, the target in the focused area absorbs the laser energy through an inverse bremsstrahlung process to vaporize and ionize, forming a high-temperature and high-pressure plasma, which then expands and explodes to remove the material [33]. The recast layer and HAZ on the sidewall are formed. As a result of the melt accumulation, protrusion is formed around the entrance to the machining area. When the high-energy short-pulse laser focuses on target materials in water, cavitation, and vaporization bubbles are generated at the machining area [34,35]. However, the bubbles, spatter, ablated materials, and vapor/plasma plume can be expelled by high-speed waterjet in CWALSM, as shown in Figure 3. With a high-speed waterjet, the ablated materials and microbubbles generated in the laser machining area could be washed away when expelled. In addition, the laser plasma generates recoil pressure on the target material during outward expansion and sputtering. If in the water region, the expansion of the plasma is confined due to the incompressibility of the water, the plasma-induced recoil pressure on the target increases [36]. Fabbro et al. [37] reported that the plasma-induced recoil pressure in the confinement of the water layer is 4–10 times higher and the duration of the shock wave is 2–3 times longer than the without water regime under the same laser power density. This tends to increase the laser ablation rate in water confinement. In addition, under the action of a high-energy short-pulse laser on the underwater target, the liquid near the solid wall heats up rapidly, and cavitation bubbles are continuously generated [38]. This is a special phenomenon of laser-matter interaction that occurs in a liquid environment. When the cavitation collapse near the solid wall, a high-speed liquid jet directed at the target is formed. When there is no water layer between the cavitation bubbles and the solid wall, liquid jet will cause a higher impulse against the wall [39]. With the action of the waterjet, the vaporization/cavitation bubbles are accelerated out of the machining area and explode. The pressure gradient caused by the explosion accelerates the plasma out of the machining area [40]. Thus, the shielding effect of the plasma plume on the incident laser energy is weakened, and more laser energy will act on the target material [15]. At the same time, the explosion of cavitation bubbles prevents the molten metal from resolidifying on the machined surface, reducing the formation of the recast layer. Finally, the excess laser energy remaining in the evaporated material is absorbed by the fluid water and taken away. Since the fluid water blocks the heat accumulation, the target material around the laser focusing area is protected, and the thermal damage is reduced. As the laser beam moves, the target material surface is rapidly cooled by the fluid water [41].

### 2.2. Optical Characteristics of Water

Due to the optical characteristics of water, there is an energy loss when the laser is transmitted in water. Therefore, the degree to which water absorbs the laser is an important factor to consider. Figure 4 illustrates that pure water has lower absorption in the green and ultraviolet bands [24,41]. The laser transmission over a 25 mm waterjet is as high as 99% at 532 nm, and 60% at 1064 nm. According to Beer-Lambert’s law, the laser intensity (*I*) after the laser beam passes through 2 mm optical quartz glass and 80 mm water layer (*l_w_*) can be calculated as:*I* = *I*_0_*T_g_*exp(−*α_w_l_w_*),(1)

Here, *I*_0_ is the laser intensity before entering the coupling cavity, *T_g_* is the transmittance of quartz glass (approximately 91%), and *α_w_* is the water absorption coefficient (0.0447 m^−1^ for 532 nm laser wavelength).

When the thickness of the water layer is 80 mm, the laser intensity (*I*) striking the workpiece surface is approximately 90.97% of the initial laser intensity (*I*_0_). The actual coupling efficiency of the laser intensity after passing through the coupling cavity is shown in Figure 5.

### 2.3. Laser-Waterjet Coupling

Before reaching the workpiece surface, the laser beam passes through three propagation media with different refractive indices: air, quartz glass, and water, as shown in Figure 6. Therefore, the refraction of the laser beam at the air-quartz glass and quartz glass-water interfaces must be considered. According to Snell’s law, the refraction angle of the laser beam in quartz glass (*θ_g_*) and water (*θ_w_*) can be determined as:(2)$$\theta_{g}{=\sin}^{-1}\left\{\frac{\sin\left[\tan^{-1}\left(r_{0}/f_{0}\right)\right]n_{a}}{n_{g}}\right\},$$
(3)$$\theta_{w}{=\sin}^{-1}\left\{\frac{\sin\left[\tan^{-1}\left(r_{0}/f_{0}\right)\right]n_{a}}{n_{w}}\right\},$$

Here, *n_a_, n_g_*, *n_w_*, *f*_0_, and *r*_0_ are the refractive index of air (1.0), the refractive index of quartz glass (1.46021), the refractive index of water (1.333), the working distance of the laser in air, and radius of the laser beam before entering the F-θ lens (2 mm), respectively. Since the refractive index of water is smaller than that of quartz glass, when the laser beam enters the water from the air, the position of the laser focus will move downward along the axis. Simultaneously, the working distance (*f*_1_) of the laser underwater will increase accordingly, expressed as:(4)$$f_{1}\;=\;L_{0}\;+\;L_{1}\;+\;L_{2}\;=\;L_{0}\;+\;L_{1}\;+\;\left[\frac{r_{0}\left(f_{0}\;-\;L_{0}\right)}{f_{0}}-{\;L}_{1}{\tan\theta }_{g}\right]{\cot\theta }_{w},$$

Here, *L*_0_ is the distance between the scanning galvanometer and the upper surface of the quartz glass, *L*_1_ is the thickness of the quartz glass, and *L*_2_ is the distance between the laser underwater working position and the lower surface of the quartz glass. At the same time, the position offset of the working point in the Z direction can be obtained:(5)$$L_{s}{=f}_{1}{-f}_{0},$$

In addition, the underwater laser focus spot diameter (*d_f_*) can be calculated as [22]:(6)$$d_{f}=\frac{{2\lambda M}^{2}}{{\pi \tan\theta }_{w}},$$

Here, *λ* is the laser wavelength; *M*^2^ is the laser beam quality.

## 3. Materials and Methods

The K4002 nickel-based special alloy with a size of 60 mm × 50 mm × 1 mm was selected for ablation and drilling experiments. Its chemical compositions, thermophysical parameters, and mechanical properties are given in Table 1, Table 2, and Table 3, respectively. Before the experiment, put the workpiece into ethanol for ultrasonic cleaning.

Figure 7 shows the experimental setup for CWALSM, which mainly includes a laser system, a scanning galvanometer, a laser-waterjet coupling system, a water supply system, and a 5-axis moving platform. It can be seen from the previous section that choosing a laser wavelength of 532 nm can reduce the loss of laser energy transmission in water. Therefore, the light source is a solid Nd: YAG nanosecond laser with a wavelength of 532 nm. The pulse width is 10.9 ns. The laser pulse frequency is 1–50 kHz. The laser average power in water is 28 W. The laser beam with a diameter of 2 mm is focused by an F-θ lens with a focal length of 160 mm. The diameter of the focused spot in the air is about 55.8 μm, and that in water is approximately 74.23 μm. The water supply system comprises an ultrapure water machine, a water pump, a pressure regulating valve, and a surge tank. The maximum water supply pressure is 0.9 MPa. In order to reduce the impact of impurities in the water on the laser transmission efficiency, ultrapure water is selected to couple with the laser beam. The diameter of the nozzle used in the experiment was 2 mm. The laser-waterjet coupling system mainly comprises quartz glass, a water cavity, and a sapphire nozzle. The laser beam enters the coupling system after passing through the F-θ lens and is transmitted in the laminar waterjet. Finally, the laser beam is focused on the workpiece surface, setting this plane as the focal plane. When it was set above or below the workpiece surface, it was considered a positive or negative focal plane position, respectively. The workpiece is fixed on the 5-axis moving platform, positioned and focused by the CCD camera.

In order to explore the effects of process parameters on the thermal damage thickness of nickel-based special alloy machined, an orthogonal experiment with five factors and four levels was designed in combination with the previous test results. Each factor and level are given in Table 4, and the process parameters are given in Table 5. The processed K4002 nickel-based special alloy samples were cut by wire-cut electrical discharge machining (WEDM). The cut specimens were ultrasonically cleaned in ethanol for 15 min. The surface profiles and microstructure observation were carried out on the Keyence VX-200 confocal laser scanning microscope (CLSM) and FEI Quanta FEG 250 scanning electron microscope (SEM). The comparison between CWALSM and laser machining in the air was carried out under the same parameters. In addition, K4002 nickel-based special alloys were percussion drilled in a single-pulse using the CWALSM to determine the ablation threshold. The experimental design parameters are listed in Table 6.

## 4. Results and Discussion

### 4.1. Ablation Threshold

The ablation threshold is related to the type of laser and material. The drilling process can only continue when the laser pulse energy exceeds the material ablation threshold. Since the area calculation method is more convenient to measure the ablation threshold and the measurement error is small, this experiment used the area calculation method to determine the ablation threshold of the nickel-based special alloy [42,43]. The prerequisite for calculating the laser ablation threshold based on the area calculation method is that the laser is a Gaussian beam. The relationship between laser fluence *φ*(*r*) and spot radius can be expressed as [43]:(7)$$\varphi \left(r\right)=\varphi_{0}exp\left(-\frac{2r^{2}}{\omega_{0}^{2}}\right),$$
where *r* is the beam section radius, *φ*_0_ is the laser peak fluence, and *ω*_0_ is the laser beam waist radius.

The relationship between laser peak fluence and laser pulse energy is:(8)$$\varphi_{0}=\frac{2E_{p}}{\pi \omega_{0}^{2}},$$
where *E_p_* is the laser pulse energy. The relationship between laser pulse energy and laser average power is:(9)$$E_{p}=\frac{P}{f},$$
where *f* is the laser pulse frequency, and *P* is the laser average power. Therefore, the relationship between the laser peak fluence at the beam waist and the laser pulse energy is:(10)$$\varphi_{0}=\frac{2P}{f\pi \omega_{0}^{2}},$$

If *φ_th_* is the laser fluence of the outer contour during damage in the area of diameter D, then *φ_th_* is the boundary energy at which the laser can damage the material, that is, the ablation threshold of the material, then [44]:(11)$$\varphi_{th}=\frac{2P_{th}}{f\pi \omega_{0}^{2}},$$
where *P_th_* is the lowest laser power to produce an effective ablation crater. From this [45]:(12)$$D^{2}=2\omega_{0}^{2}\ln\left(\frac{\varphi_{0}}{\varphi_{th}}\right),$$

Bringing Equations (10) and (11) into Equation (12), can obtain:(13)$$D^{2}=2\omega_{0}^{2}\ln\left(P\right)-2\omega_{0}^{2}\ln\left(P_{th}\right),$$

Equation (13) can be regarded as a linear relationship of *D*^2^-ln*P*, with a slope of 2*ω*_0_^2^. Therefore, according to the definition of laser ablation threshold, the ablation threshold can be obtained by the *D*^2^ vs. ln*P* curve with an extrapolation to *D*^2^ = 0. By sorting out and calculating the experimental data, the relationship between the square of the ablation diameter and the logarithm of the laser average power (waterjet speed of 5.19 m/s) was obtained. The results are shown in Figure 8. It can be seen from Figure 8 that the diameter of the laser single pulse ablation crater increases with the increase of the laser average power. By calculating the fitting curve *D*^2^-ln*P*, when the ablation craters diameter *D*^2^ = 0, it can be concluded that the K4002 nickel-based special alloy coaxial waterjet assisted laser single pulse ablation threshold is *φ_th_* = 4.15 J/cm^2^.

### 4.2. Variance and Range Analysis

Laser machining is a multi-parameter coupled nonlinear process. It is the key to laser machining to explore the effects of each parameter on the thermal damage thickness layer under the coupling effect. In this paper, the variance and range analysis were used to process the orthogonal experiment data, and the effects of various factors on the thermal damage thickness and the optimal level combination were obtained. The thermal damage thickness obtained in each group of tests is shown in Figure 9. As can be seen from the figure, the maximum thermal damage layer thickness is 2.775 μm. In particular experimental conditions, CWALSM can realize machining without thermal damage.

According to the thermal damage thickness, the variance and range analysis of the test are carried out. The analysis results are shown in Figure 10 and Table 7.

It can be seen from Figure 10 that when the laser pulse energy is used as a factor, and the laser pulse frequency, pulse overlap, focal plane position, and waterjet speed are used as covariates, the *p*-value of the laser pulse energy is greater than 0.05, indicating that the laser pulse energy has no significant effect on the thermal damage thickness. When laser pulse frequency is used as a factor, and laser pulse energy, pulse overlap, focal plane position, and waterjet speed are used as covariates, the *p*-value for laser pulse frequency is 0.011, which is less than 0.05, indicating that laser pulse frequency has a significant effect on thermal damage thickness. When the pulse overlap is used as a factor, and the laser pulse energy, laser pulse frequency, focal plane position, and waterjet speed are used as covariates, the *p*-value of the pulse overlap is less than 0.001, indicating that the impact of the pulse overlap on the thermal damage thickness is highly significant. When the focal plane position is used as a factor, and laser pulse energy, laser pulse frequency, pulse overlap, and waterjet speed are used as covariates, the *p*-value of focal plane position is greater than 0.05, indicating that the effect of focal plane position on thermal damage thickness is not significant. When the waterjet speed is used as a factor, and the laser pulse energy, laser pulse frequency, pulse overlap, and focal plane position are used as covariates, the *p*-value of the waterjet speed is 0.062, which is closer to 0.05, indicating that the effect of the waterjet speed on the thermal damage thickness is significant.

The mean value of each factor at each level was calculated, and the effect of each factor on thermal damage thickness was obtained by calculating the range R_J_ of each factor. The factors affecting the thermal damage thickness were sorted according to the magnitude of the range, as shown in Table 6. The most effective factor is laser pulse frequency (R_J_ = 1.7858), followed by waterjet speed (R_J_ = 1.2898), pulse overlap (R_J_ = 1.2260), laser pulse energy (R_J_ = 0.4083), and focal plane position (R_J_ = 0.1153). Of these, the laser pulse frequency has the greatest effect on the thermal damage thickness and is the main factor affecting the thermal damage thickness. The waterjet speed and pulse overlap have little effect on the thermal damage thickness. Laser pulse energy and focal plane position have the least effect on thermal damage thickness and are secondary factors affecting thermal damage thickness.

In addition, the main effect curve of various factors on the thermal damage thickness was obtained, as shown in Figure 11. It can be seen from the figure that the regression line at the focal plane position is relatively gentle, and has no significant effect on the thickness of the thermally damaged. The regression line of laser pulse frequency and waterjet speed is steep, which significantly impacts thermal damage thickness, and reducing laser pulse frequency or increasing waterjet speed can reduce thermal damage thickness. This is because most of the heat in the machining area is absorbed by water during the laser pulse-off time, which reduces the heat transfer and accumulation of the material [46]. At the same time, the high-speed waterjet with a stronger cooling effect and initial kinetic energy can take away the ablated material in the machining area, reducing the formation of the recast layer. When the laser pulse frequency is 1 kHz, the thermal damaged thickness is minimum. However, when the laser pulse frequency is too high, similar to continuous laser machining, the cooling effect of the waterjet is weakened. Figure 11 also shows that as the laser pulse energy increases, the heat effect cannot be offset by the cooling effect from waterjet flushing and heat diffusion [32].

Accordingly, the optimal level combination of various factors for the minimum thermal damage thickness can be preliminarily selected as laser pulse energy 0.21 mJ, laser pulse frequency 1 kHz, pulse overlap 55%, focal plane position 1.0 mm, and waterjet speed 6.98 m/s. The above parameters were used to test, and it was observed by SEM that the cut edges are straight. Furthermore, the cut sides are clean. There is no plastic deformation or thermal damage found. Figure 12 illustrates the microstructure of the cross-sectional profile of the hole middle.

### 4.3. Comparison between CWALSM and Laser Machining in Air

In order to highlight the advantages of CWALSM in machining quality and low thermal damage, laser ablation in air and coaxial waterjet-assisted laser ablation were compared under the corresponding condition. The laser pulse energy of 0.53 mJ, laser pulse frequency of 15 kHz, pulse overlap of 85%, the focal plane position of −1.0 mm, and waterjet speed of 6.98 m/s were applied. Figure 13 and Figure 14 are the 3D and cross-section profiles of the holes obtained from the laser machining in air and CWALSM. Figure 15 and Figure 16 show the hole entrance and exit morphologies in air and CWALSM. It can be seen from the comparison of Figure 13 and Figure 14 that when laser machining is carried out in the air, obvious accumulation is formed around the entrance of the machining area, resulting in a poor 3D profile and cross-sectional quality of the machined surface. In addition, there is significant spatter and recast material at the edge of the hole entrance, as shown in Figure 15a. In contrast, the edge of the hole exit is clean and tidy, and only a small amount of spatter and recast material can be observed through the locally enlarged view, as shown in Figure 16a. Because under the action of a nanosecond short-pulse laser, when the surface temperature of the substrate material is close to its thermodynamic adjacent temperature, the material changes from a superheated liquid to a mixture of vapor and liquid droplets. Due to the rapid heat conduction of the nickel-based superalloy, the substrate material adjacent to the laser irradiation area solidifies immediately after melting to form a recast layer. At the end of the laser pulse, molten droplets are ejected at supersonic speeds, creating a dense plasma with high temperature and pressure. Part of the ejected molten droplets adheres to the side wall and solidifies to form a recast layer. The other part is re-solidified into a thin film, thereby changing the morphology of the edge of the ablated area and the surrounding area [47]. A different substance from the substrate was observed in the SEM images, which may have been slightly contaminated before the sample was observed, but did not affect the observation.

In contrast, no protrusion is formed with CWALSM. The transition between the machined edge and the unmachined surface is smooth, as shown in Figure 14. Figure 15b and Figure 16b show that the recast layer is formed on the surface of the machined edge due to the solidification of the melt. These melts were spread on the material surface from waterjet flushing. In addition, micro pits formed under the action of microjets produced by cavitation/microbubble explosions. When the cavitation bubbles and microbubbles formed in the machining area are washed by the waterjet, the high temperature and high-speed jet are formed due to the pressure gradient between the adjacent material surface. Under the confinement of the water, the jet exerts an impact force on the material surface that is of the same magnitude as the laser ablation force [48], causing damage to the material surface. This damage has less effect on the hole exit. It is found that the Ra of the micro pits around the hole entrance is close to the substrate material when measured at a multiple of ×200. In addition, due to the distribution of laser energy on the substrate material and the shielding effect of the plasma plume on the laser beam, the formation of a taper angle in laser machining is inevitable [11,32]. The taper angle of the machined holes was reduced by 106%, with a taper angle of 3.38° and 1.64° of holes obtained by laser machining in air and CWALSM, respectively.

Since laser machining is a thermal machining process, thermal damage (such as recast layer and HAZ) is difficult to avoid during machining. The heat generated in laser machining continuously diffuses to the surrounding materials. Although the surrounding materials are not melted, the microstructure changes.

For nanosecond pulse laser machining in air, the interaction process between laser and material is dominated by heat conduction, and materials are removed by melting/vaporization. During the pulse-off time, molten material will build up and re-solidify on the hole sidewalls, which tends to create a recast layer. The boundary of thermal damage is specified where the metallurgical structure of the substrate material is changed from the original one to the otherwise. Figure 17 shows the cross-sectional structures of the hole drilled by laser machining in the air. As shown in Figure 17, it can be found that the thermal damage is the largest at the hole entrance and decreases with the increasing distance from the entrance. As the machining depth increases, the plasma recoil pressure is not enough to expel all the ablated material out of the machining area. Under the repeated action of the laser, the thermal damage in the machining area increases. When the hole is penetrated, the recast material attached to the hole side wall absorbs heat and re-melts/vaporizes is expelled. The plasma/ablation particles have an escape channel, which reduces the thermal damage to the hole side wall. The thermal damage thickness on the hole side wall is not uniform, and the thermal damage thickness range caused by laser machining in the air is about 0.499–2.394 μm. In addition, the recast layers are scaly and with the typical features of the material that melts and re-solidifies[49], as shown in Figure 17. In addition, there are microcracks between the recast layer and the substrate material.

Figure 18 shows the cross-sectional structures of the hole drilled by CWALSM with a waterjet speed of 6.98 m/s. Compared with laser machining in air, the thermal damage of each part is significantly reduced, and the thickness range is about 0–0.966 μm. Furthermore, the hole-side wall machining texture is clear at the exit, and no apparent thermal damage can be observed. It shows that CWALSM significantly reduces the thermal effect of the laser on the substrate material around the machining area. CWALSM can avoid thermal damage mainly due to the influence of fluid water. Because the fluid water can block heat build-up, the material around the laser focus area is protected from thermal damage. As laser machining proceeds, the substrate material surface is rapidly cooled by fluid water. Furthermore, it can be observed that the hole side wall is not neat, and it is jagged from the entrance to the middle part, as shown in Figure 18. However, this phenomenon disappears at the hole exit. This will affect the roughness of the hole sidewalls. The reason for this phenomenon may be related to the laser pulse frequency. When the laser pulse frequency increases, the inter-pulse interval shortens. When the waterjet speed is constant, the action of the previous laser pulse has not yet ended, and the next laser pulse will act on the cavitation bubbles or plasma that are too late to be expelled, causing the laser beam to refract and cause damage to the hole wall. The reason for the weak effect on the hole exit is that after the hole is pierced, the cavitation bubbles and plasma have an expelled channel, which weakens the refraction of the laser beam.

## 5. Conclusions

In the present research, single-pulse ablation and drilling experiments were carried out on K4002 nickel-based special alloy by CWALSM. The effects of laser pulse energy, laser pulse frequency, pulse overlap, focal plane position, and waterjet speed on the thermal damage thickness were studied by orthogonal experiments. Some conclusions can be drawn as follows:

(1) The ability of CWALSM to ablate nickel-based special alloys was studied. The nanosecond laser single pulse ablation threshold of K4002 nickel-based special alloy was determined to be 4.15 J/cm^2^ by the area calculation method.

(2) The influence of each process parameter on the thermal damage thickness from high to low is as follows: laser pulse frequency, waterjet speed, pulse overlap, laser pulse energy, and focal plane position. In addition, the optimized experimental parameters obtained by the range analysis of the orthogonal experiment are as follows: laser pulse energy 0.21 mJ, laser pulse frequency 1 kHz, pulse overlap 55%, focal plane position 1.0 mm, waterjet speed 6.98 m/s. Under this process parameter, heat-damage-free machining can be realized.

(3) The 3D and cross-sectional profiles of the hole were investigated. Compared with laser machining in the air with the same parameters, the taper angle can be decreased by 106% on the holes processed by CWALSM. There is no protrusion at the hole entrance, and the texture of the hole side wall is clear. The thermal damage in the laser-material interaction area is significantly reduced.

(4) The thermal damage thickness in the machining area was investigated. The thermal damage thickness is the largest at the hole entrance and decreases with the distance increase from the entrance. The thermal damage thickness range is 0.499–2.394 μm by laser machining in air, and there is an obvious recast layer. In contrast, the thermal damage thickness range is 0–0.996 μm by CWALSM.

## Figures and Tables

**Figure 1 micromachines-14-00641-f001:**
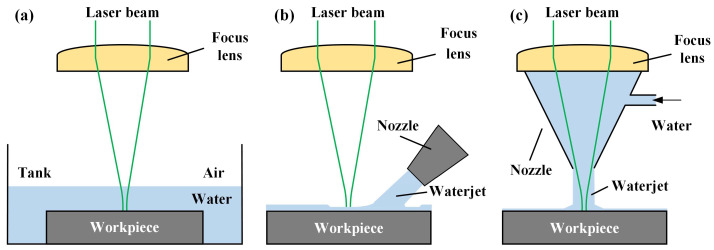
Schematic of water-assisted laser machining. (**a**) underwater laser machining; (**b**) off-axis waterjet assisted laser machining; (**c**) coaxial waterjet assisted laser machining.

**Figure 2 micromachines-14-00641-f002:**
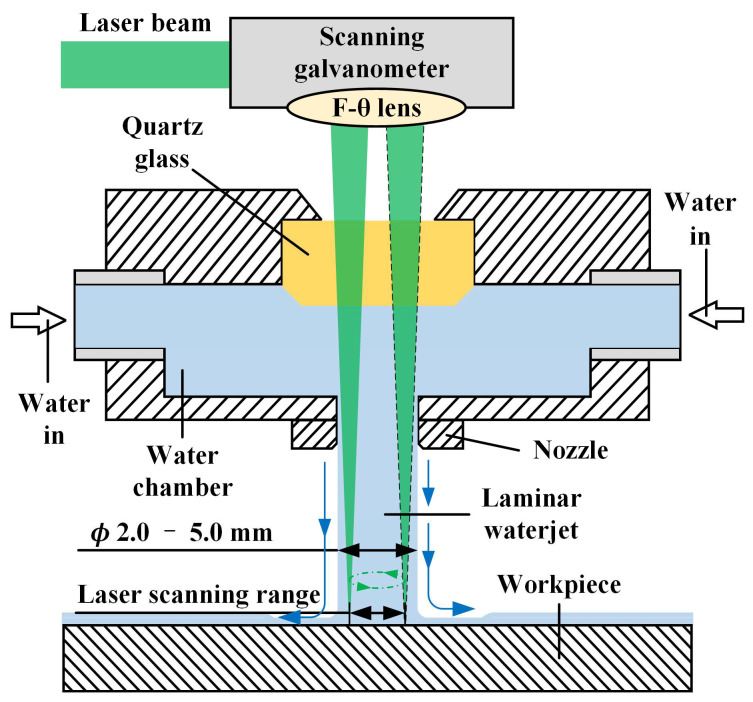
Working principle of coaxial waterjet assisted laser scanning machining (CWALSM).

**Figure 3 micromachines-14-00641-f003:**
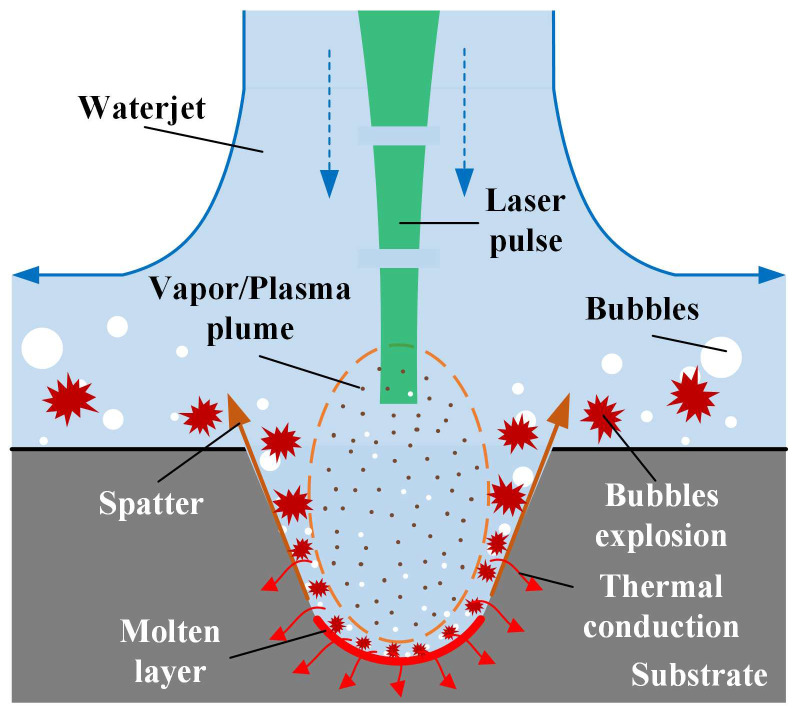
Working principle of coaxial waterjet assisted laser scanning machining (CWALSM).

**Figure 4 micromachines-14-00641-f004:**
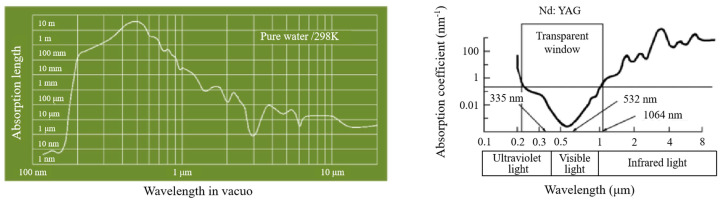
Absorption lengths of pure water to the laser of different wavelengths.

**Figure 5 micromachines-14-00641-f005:**
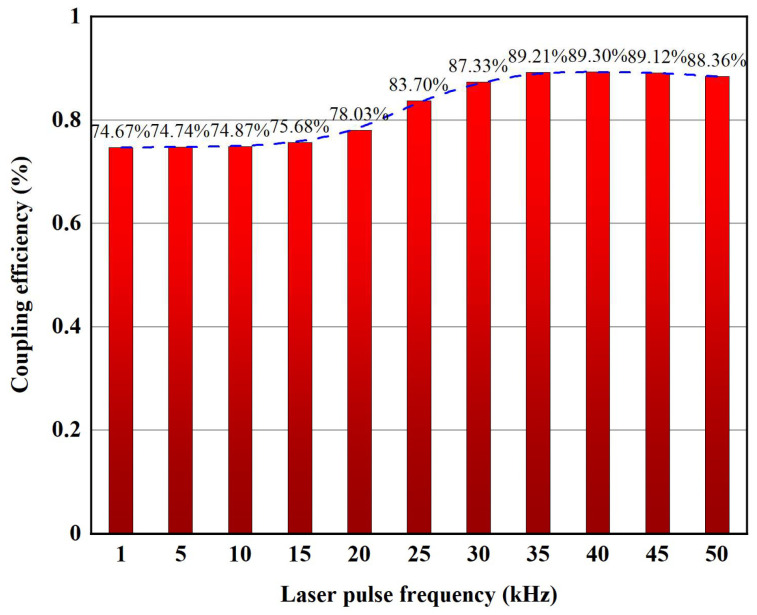
The coupling efficiency of the laser and waterjet.

**Figure 6 micromachines-14-00641-f006:**
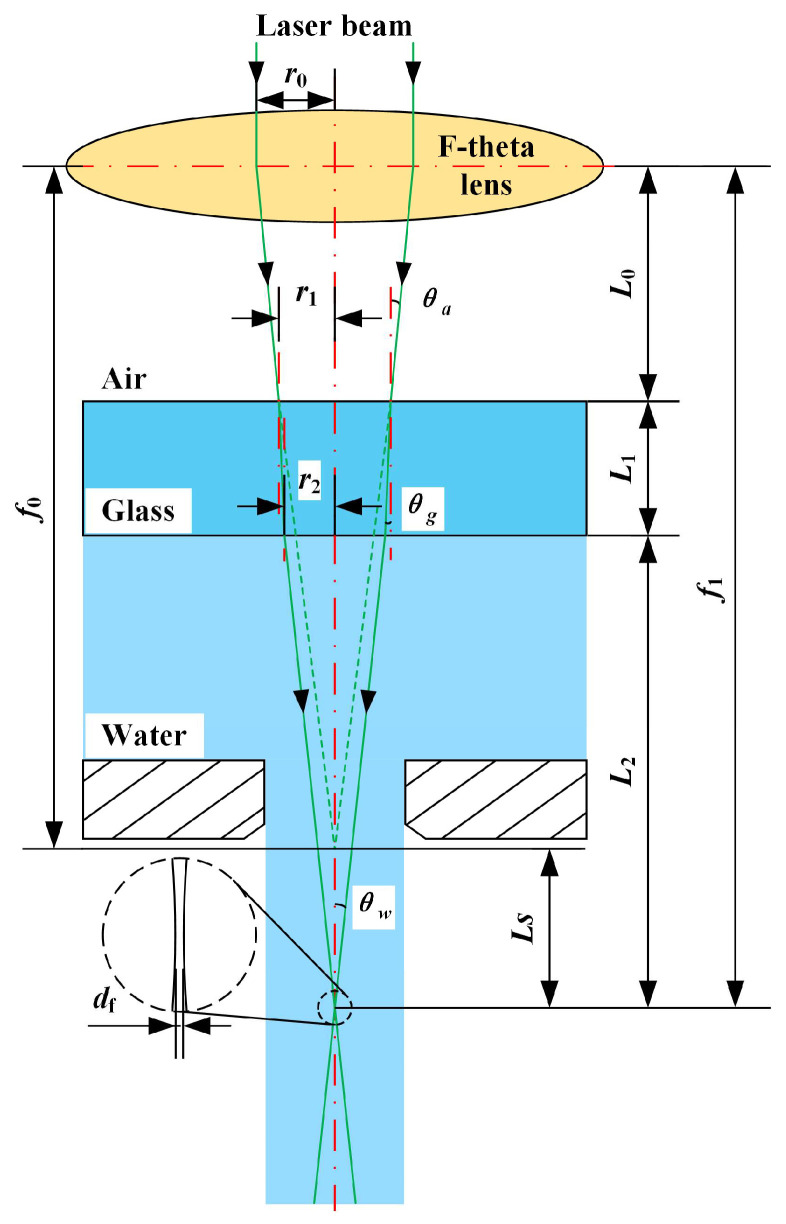
Refraction of the laser beam in quartz glass and water.

**Figure 7 micromachines-14-00641-f007:**
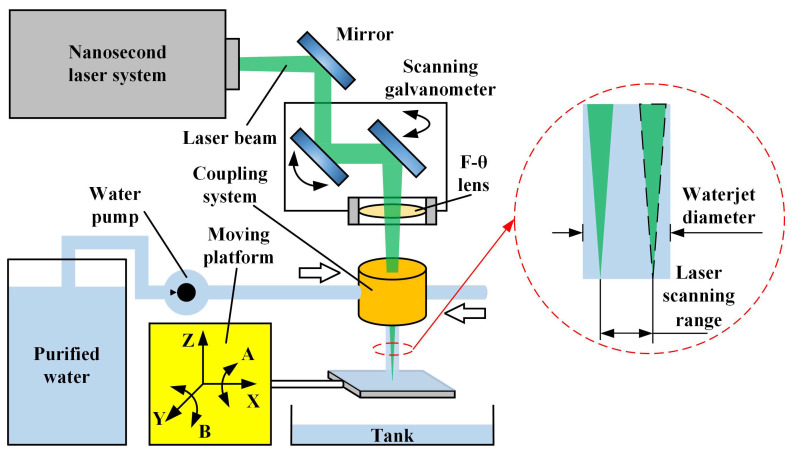
Experimental setup for CWALSM.

**Figure 8 micromachines-14-00641-f008:**
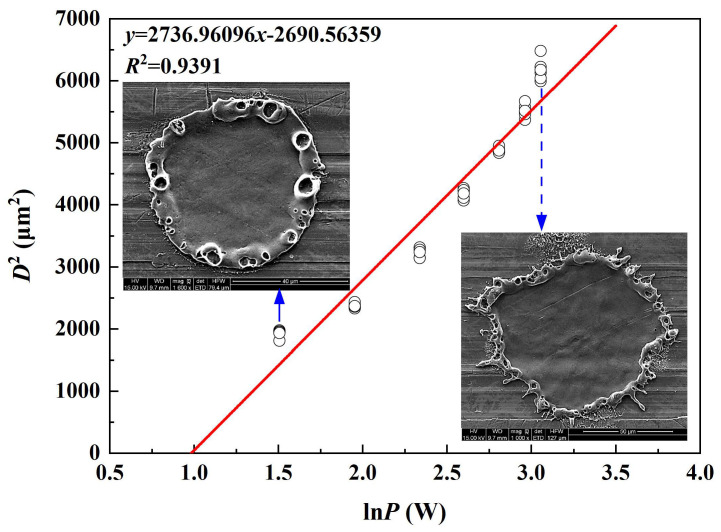
Squared diameters *D*^2^ of the ablated craters are correlated with ln*P*.

**Figure 9 micromachines-14-00641-f009:**
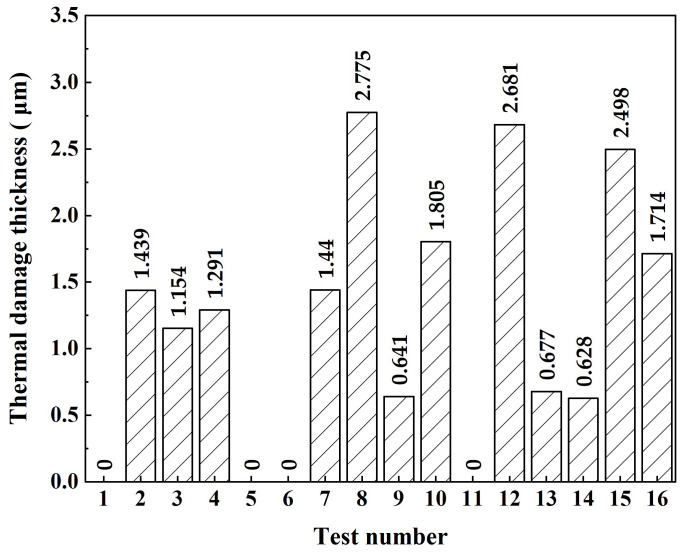
The thermal damage thickness was obtained in each experimental group.

**Figure 10 micromachines-14-00641-f010:**
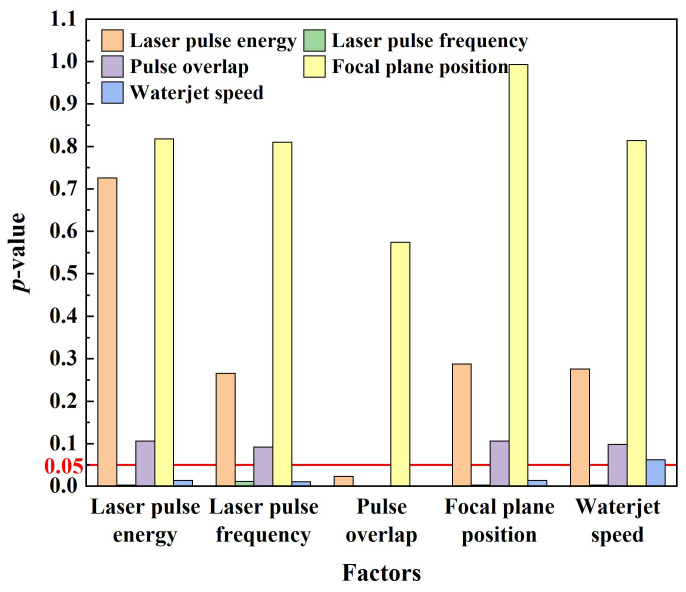
Variance analysis diagram of orthogonal experiment.

**Figure 11 micromachines-14-00641-f011:**
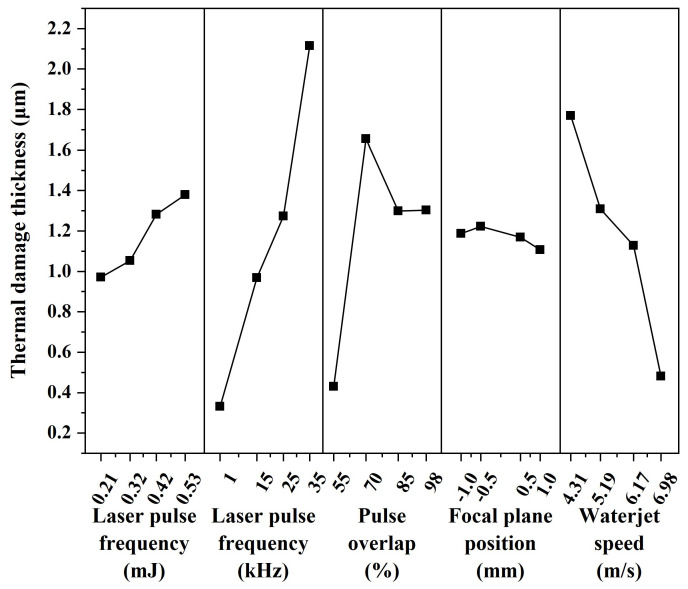
Main effect curve diagram.

**Figure 12 micromachines-14-00641-f012:**
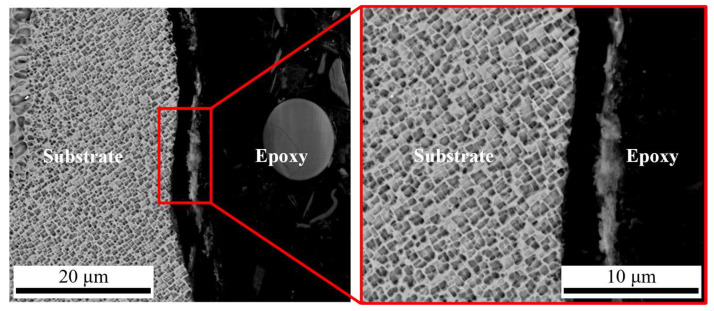
SEM images of CWALSM at optimum parameters level.

**Figure 13 micromachines-14-00641-f013:**
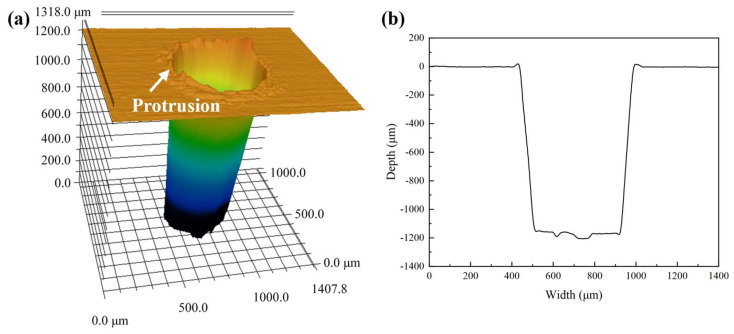
Hole machined by laser in air. (**a**) 3D profile, (**b**) Cross-sectional profile.

**Figure 14 micromachines-14-00641-f014:**
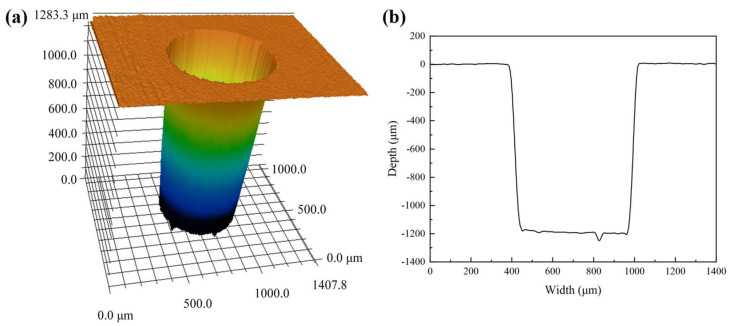
Hole machined by CWALSM. (**a**) 3D profile; (**b**) Cross-sectional profile.

**Figure 15 micromachines-14-00641-f015:**
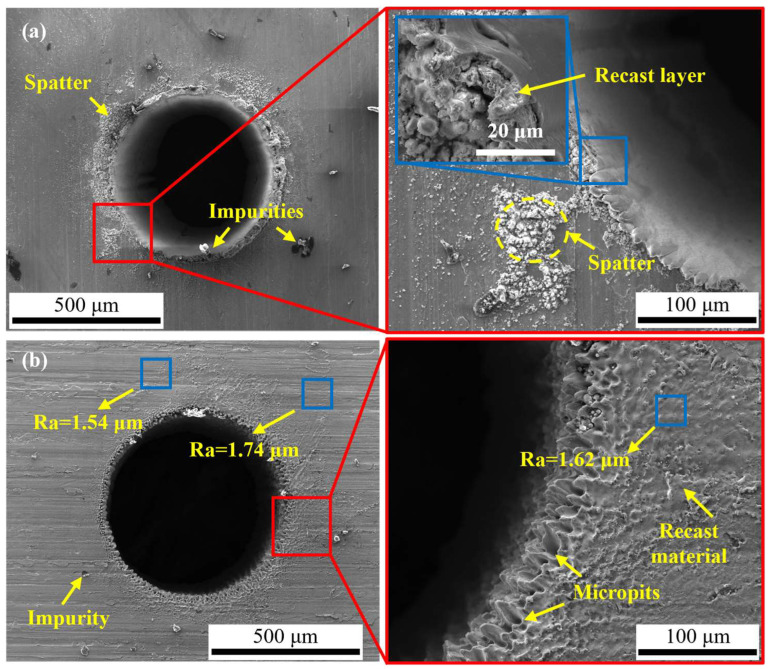
SEM morphologies of the hole entrance by laser machining in different conditions. (**a**) Laser machining in the air; (**b**) CWALSM.

**Figure 16 micromachines-14-00641-f016:**
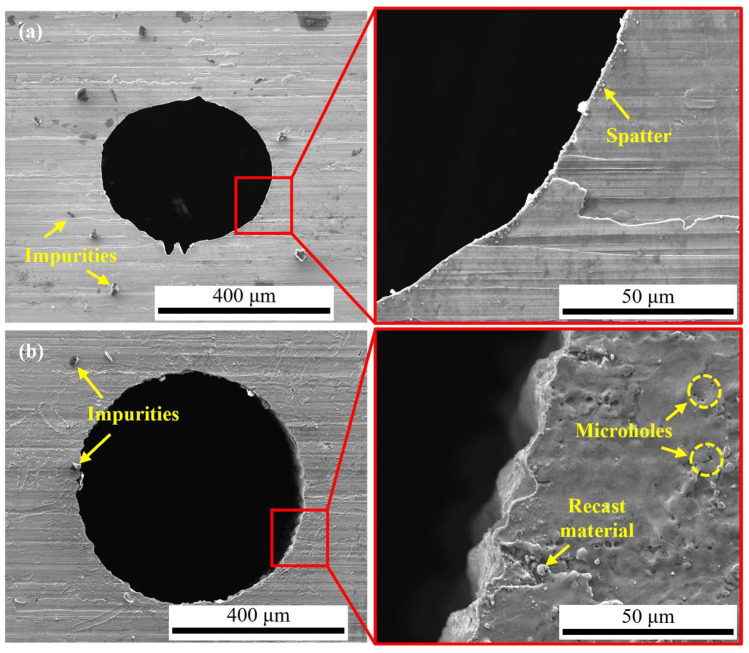
SEM morphologies of the hole exit by laser machining in different conditions. (**a**) Laser machining in the air; (**b**) CWALSM.

**Figure 17 micromachines-14-00641-f017:**
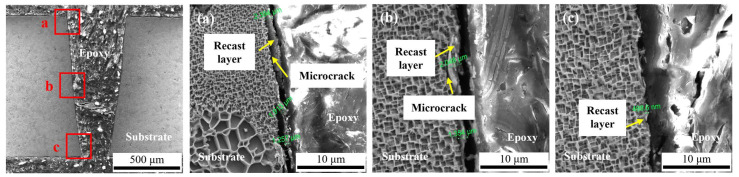
SEM morphologies of the hole machined by laser machining in air. (**a**) Hole entrance; (**b**) Hole middle; (**c**) Hole exit.

**Figure 18 micromachines-14-00641-f018:**
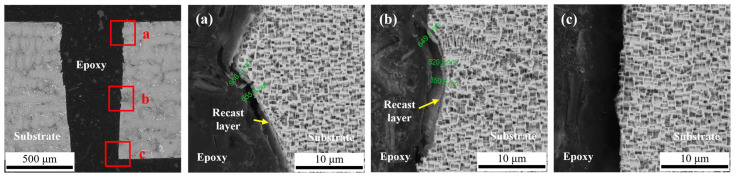
SEM morphologies of the hole machined by CWALSM. (**a**) Hole entrance; (**b**) Hole middle; (**c**) Hole exit.

**Table 1 micromachines-14-00641-t001:** Chemical compositions of K4002 nickel-based special alloy.

Element	C	Cr	Co	W	Al	Ti	Ta	Hf	B	Zr	Ni
Wt. [%]	0.15	9	10	10	5.5	1.5	2.5	1.45	0.015	0.055	Bal.

**Table 2 micromachines-14-00641-t002:** Thermophysical parameters of K4002 nickel-based special alloy.

Physical Properties	Values
Density [g∙cm^−3^]	8.5
Specific heat [J∙kg^−1^∙°C^−1^]	419–595
Thermal conductivity [W∙m^−1^∙°C^−1^]	7.54–22.19
Melting point [°C]	1280–1380
Thermal diffusivity [10^−6^∙m^2^∙s^−1^]	2.1–4.4

**Table 3 micromachines-14-00641-t003:** Mechanical properties of K4002 nickel-based special alloy.

Mechanical Properties	Values
Yield strength *R_p_*_0.2_ [MPa]	737 (≥)
Tensile strength *R_m_* [MPa]	687 (≥)
Elongation *A* [%]	32
Reduction in cross section on fracture *Z* [%]	23
HBW	143

**Table 4 micromachines-14-00641-t004:** Configuration of experiment parameters.

Factor	Level			
	1	2	3	4
Laser pulse energy [mJ]	0.21	0.32	0.42	0.53
Laser pulse frequency [kHz]	1	15	25	35
Pulse overlap [%]	55	70	85	98
Focal plane position [mm]	−1.0	−0.5	0.5	1.0
Waterjet speed [m/s]	4.31	5.19	6.17	6.98

**Table 5 micromachines-14-00641-t005:** Orthogonal experimental design.

Number	Laser Pulse Energy [mJ]	Laser Pulse Frequency [kHz]	Pulse Overlap [%]	Focal Plane Position [mm]	Waterjet Speed [m/s]
1	0.21	1	55	−1.0	4.31
2	0.21	15	70	−0.5	5.19
3	0.21	25	85	0.5	6.17
4	0.21	35	98	1.0	6.98
5	0.32	1	70	0.5	6.98
6	0.32	15	55	1.0	6.17
7	0.32	25	98	−1.0	5.19
8	0.32	35	85	−0.5	4.31
9	0.42	1	85	1.0	5.19
10	0.42	15	98	0.5	4.31
11	0.42	25	55	−0.5	6.98
12	0.42	35	70	−1.0	6.17
13	0.53	1	98	−0.5	6.17
14	0.53	15	85	−1.0	6.98
15	0.53	25	70	1.0	4.31
16	0.53	35	55	0.5	5.19

**Table 6 micromachines-14-00641-t006:** Main parameters used in ablation threshold experiment.

Laser Pulse Frequency [kHz]	Number of Pulses	Laser Power [W]
30	1	4.51, 7.05, 10.35, 13.46, 16.59, 19.34, 21.21

**Table 7 micromachines-14-00641-t007:** Range analysis of the orthogonal experiment.

Range	Factor				
	Laser Pulse Energy [mJ]	Laser Pulse Frequency [kHz]	Pulse Overlap [%]	Focal Plane Position [mm]	Waterjet Speed [m/s]
*k* _J1_	0.9710	0.3295	0.4285	1.1873	1.7695
*k* _J2_	1.0537	0.9680	1.6545	1.2228	1.3085
*k* _J3_	1.2817	1.2730	1.2995	1.1683	1.1280
*k* _J4_	1.3793	2.1153	1.3032	1.1075	0.4798
R_J_	0.4083	1.7858	1.2260	0.1153	1.2898
Rank	4	1	3	5	2

## Data Availability

Not applicable.

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
