# Peer review of "Experimental Study on Coaxial Waterjet-Assisted Laser Scanning Machining of Nickel-Based Special Alloy"

_micromachines, 2023, doi:10.3390/mi14030641_

Round 1

Reviewer 1 Report

This paper adopts laser scanning underwater processing method, which has its novelty. However, the research content does not highlight the advantages and disadvantages of this method in depth. The advantages and disadvantages of this method with underwater laser processing and water guided laser processing should be specifically analyzed. Therefore, the mechanism research in section 2.1 should be more focused on the more in-depth mechanism analysis of this method. The experimental results should not be limited to comparison with air, but more valuable compared with underwater or water guided methods.

Reviewer 2 Report

The authors presented effects of Coaxial Waterjet Assisted Laser Scanning Machining (CWALSM) for reducing the conduction and accumulation of heat in laser machining by the high specific heat capacity of water. For the experiment, a nickel-based superalloy K4002 was used. The laser ablation threshold observed and slight thermal damage in the machining area.

The article is very interesting and good fits the profile of the journal. I have no fundamental objections. I think it will be suitable for publication after corrections.

Strength

1.       The strength of this paper is a very wide analysis of the state of the issue.

Weak

1.       The authors, in my opinion, mistakenly call the special alloy K4002 by the name of super alloy. This is a mistake, as even the manufacturer/distributor (https://www.superalloys.net/) confirms this. This should be changed in the title and body of the paper.

Noticed errors

1.    Please clearly highlight the research gap observed from the literature analysis.

2.    Line 166. Please do not use unnecessary units. Since most of the paper uses mm, so be it. Please convert cm to mm (1 cm = 10 mm).

3.    In the chapter 3. Materials and Methods lacks mechanical properties of K4002 special alloy.

4.    Table 2. The multiplication operator is the sign: · , not . Please correct this throughout the paper.  

Small errors

1.       Line 43. Is: 1.2mm; should be: 1.2 mm. Please correct this throughout the paper

2.       Line 136. Is: Φ2mm~Φ5mm; should be: ø2 mm ÷ ø5 mm.

3.       Pages 3/4, 4/5, 6/7, 8/9, 15/16, 16/17. Bad pagination.

4.       Equations numbers (4) and (5) are in italics. They should be a regular font. 

Round 2

Reviewer 1 Report

 Accept in present form